# Relaxation in the Chinese Hukou System: Effects on Psychosocial Wellbeing of Children Affected by Migration

**DOI:** 10.3390/ijerph16193744

**Published:** 2019-10-04

**Authors:** Jingjing Lu, Minmin Jiang, Lu Li, Therese Hesketh

**Affiliations:** 1Department of Social Medicine, School of Public Health, Zhejiang University, Hangzhou 310027, China; jingjinglu@zju.edu.cn (J.L.); jiangm@zjnu.edu.cn (M.J.); Lilu@zju.edu.cn (L.L.); 2Centre for Global Health, School of Medicine, Zhejiang University, Hangzhou 310027, China; 3Institute for Global Health, University College London, London WC1N1EH, UK

**Keywords:** migration, China, children, psychological well-being

## Abstract

The hukou household registration system in China is being relaxed in small and medium-sized cities, which makes it easier for migrant worker parents to take their children with them to cities for work. The aim of this study was determine the potential impact on psychological well-being of this change for children by migration and hukou status. We conducted a cross-sectional survey using the Chinese version of the Strengths and Difficulties Questionnaire (SDQ) in urban and migrant schools in the capital, Hangzhou, and in schools in two rural counties of Zhejiang Province. Completed questionnaires were obtained from 2965 children, aged 10 to 15: 867 urban children with urban hukou, 625 migrant children with rural hukou, 695 rural children with rural hukou living with parents, and 778 left behind children. The crude SDQ scores showed that children directly affected by migration (migrant children and left behind children) were most at risk of psychological problems; urban and rural children living with their parents in their accustomed environment were least vulnerable. After adjustment for confounders, migrant children were the most vulnerable to psychological problems. Given that migration with children is on the increase, efforts should be made to improve conditions in urban areas for these children, and especially to ensure access to urban public schools.

## 1. Introduction

Patterns of internal migration in China are changing, as China progresses towards a target of 70% urbanisation by 2030, from the current rate of 57% [1]. Urbanisation is seen as essential to China’s continued economic growth, as a driver of productivity and consumption [2]. Urbanisation has steadily increased since the early 1980s, when rural-urban migration for work started, as a means of providing cheap labour in the new and rapidly developing manufacturing sector, which would fuel the Chinese economic boom [1]. In 2017, migrant workers numbered an estimated 288 million, equivalent to one-third of the entire working population [3].

Chinese internal migration has always been controlled by the household registration or hukou system. This system is used by the government to regulate population distribution within the country, the urbanisation of cities, and the equality of urban and rural residents’ status [4]. All Chinese citizens have a hukou, which acts like an internal passport and allows access to local social benefits, including education and healthcare. It is mostly allocated by place of birth, and possession of an urban hukou, especially for a larger city, is regarded as highly desirable.

Until recently, migrant workers, who by definition lack urban hukou, have been prevented from accessing such benefits without additional payment. They have also been prevented from obtaining urban hukou, irrespective of their length of stay [4]. In addition, the hukou system, to some extent, perpetuates negative social attitudes and prejudices that urban residents often hold for rural residents. Importantly, the hukou system has influenced patterns of family migration. Initially, single people, predominantly men, dominated migration. However, by the mid-1990s many migrants were joined by their partners [5]. Strict hukou regulations created barriers to services, and together with the often precarious nature of migrant life, this led most migrants to leave their children behind with family members [5]. However, the hukou system is gradually being relaxed, especially in small and medium-sized cities [6]. This is being driven by the urbanisation imperative (consistent with the 2016 goal, of bringing 100 million people into cities by 2020), the need *to solve migrant workers’ residency challenges*, and the desire to boost domestic consumption [6]. In small and some medium-sized cities (defined as a population of up to five million), hukou conversion from rural to urban is actually being encouraged [7]. Some bigger cities are also relaxing restrictions, allowing for better access to services for migrant workers, and improving overall living conditions. In contrast, the largest cities, notably Beijing, Shanghai, Shenzhen, and Guangdong, which tend to be magnets for migrant workers, still have restrictive hukou policies and have instituted population caps for permanent residents [8]. Accordingly, the government is relaxing residency restrictions in smaller cities, such as Huzhou and Jiangshan in Zhejiang Province, while concurrently restricting them in larger ones [9].

These changes in the hukou system have implications for migrant workers and their families. Around 100 million children, or 36% of all Chinese children under 17, are directly affected by internal migration, either through migrating with parents or being left behind by migrant parents [10]. The relaxations in the hukou system are leading to changes in the relative proportions of these groups, with more migrants now bringing their children with them to cities, and fewer leaving their children behind. In 2016, it was estimated that around 80% of new migrants were accompanied by their partners, and over 50% of migrant parents by their children [3]. From 2012 to 2017, estimates for numbers of migrant children increased from 32 to 36 million, while the numbers of left behind children (LBC) declined by four million to around 60 million across the country [3].

Despite the changes in the hukou system migrant worker parents still face a difficult decision: to separate from their children, leaving them behind to be cared for by relatives, or to take their children with them to a new environment [11]. Either might have detrimental effects for parents and children. There is a wealth of literature describing how these detrimental effects might manifest for left behind children, including increased risk of depression, anxiety, suicidal ideation, and conduct disorders [12,13]. Some studies have identified exacerbating factors, such as poverty and parental divorce and mitigating factors, such as good regular communication with parents, and good quality of substitute parenting [14]. Less attention has been paid to the migrant children who accompany their parents: they may benefit from living with their parents, but many migrants’ long working hours limit the time that they can spend with their children, and a lack of family support can exacerbate the situation [15]. Hukou restrictions have long been blamed for pushing rural migrant workers to the margins of urban society. In many larger cities, migrant children are prevented from enrolling in public schools, so they attend migrant schools, which typically have poor facilities and unqualified teachers [16]

Against the background of the changes in the hukou system, parents may now find the decision about taking their children even more difficult. Therefore, the effects of staying behind or migrating with parents on the well-being of children is an important comparison. The aim of this study is to compare psychological well-being in children, by migration and hukou status, in Zhejiang Province, an eastern coastal province with a population of 57 million. It is both a feeder and receiving province for migrant workers, with over 10 million migrant workers, 1.4 million left behind children, and nearly a million urban migrant children [17]. Hukou reform is underway in a number of its cities, with easier access to services for migrants. The numbers of migrant children are expected to increase and left behind children to decrease over the next decade, so Zhejiang presents a good example of expected patterns of migration across China for the future. Four groups of children were compared: urban children with urban hukou, migrant children with rural hukou living with parents in cities, rural children with rural hukou living with parents, and LBC with rural hukou.

## 2. Materials and Methods

### 2.1. Setting and Sample

We conducted a cross-sectional study in Zhejiang Province, from March 2016 to March 2017. The participants were recruited through a school-based sampling procedure. To recruit urban and migrant children, two migrant primary schools, two migrant middle schools, two public primary schools, and two public middle schools in the capital city Hangzhou, were randomly selected from the school roster of the City Education Bureau. One class per year group, Years 5–6 in each primary school and Years 7–8 in each middle school, was randomly allocated to participate in the study. The age range of these years is 10–15, and it was selected to allow for self-completion of questionnaires, and because the SDQ is regarded as an especially appropriate tool for use in this age group [18]. Rural and left-behind children were recruited through a school-based multistage sampling procedure. Two rural counties with at least 35% left behind children and varying economic status were selected. In each county, three primary schools and three middle schools were randomly selected from a complete list of all schools in the county. One random class per year group, Years 5–6 in primary school and Years 7–8 in middle school, was randomly allocated to participate in the study.

### 2.2. Procedure

Study information was sent to the local Education Bureaus and the head teachers of each school. They all agreed to participate and gave formal approval. A standardized mode of administration of the questionnaire was carried-out across all schools. An information letter and a consent form were distributed to parents and guardians. The questionnaire was only distributed to students where guardian consent had been provided. Research assistants approached the children in the classroom, explained the purpose of the questionnaire, told them that everything they wrote was anonymous and confidential, obtained their verbal consent, and offered to help if there were any queries. After completion, all of the questionnaires were placed into a box that was sealed.

The study was approved by the Ethics Committees of UCL and Zhejiang University (Ref no. ZGL201412-2). We were very aware of the sensitivity of some of the questions and so at the end of the questionnaire a hotline number for an adolescent mental health service was provided.

### 2.3. Measures

Socio-demographic characteristics included: age, gender, migrant status, family’s economic status, and parents’ education level. Special care was taken to ensure accuracy of migration status. We only included children who had been left behind for at least a year, and migrant children who had been living at the destination for at least a year, to allow for issues of acclimatization in both instances, and allow for more valid comparison. Accordingly, as an additional check, children were verbally asked about the duration of absence of both parents for LBC, or the duration at the destination for migrant children. Family economic status was measured by possession of certain household items: air conditioner, refrigerator, washing machine, computer, and private car. This was then coded as low (0–2 items), moderate (3–4 items), and high-income (five items). Parents’ education level referred to the higher education level of the two parents.

Psycho-social wellbeing was measured with the Chinese version of the self-reported version of the Strengths and Difficulties Questionnaire (SDQ-C) [18]. The English version of the SDQ has been very widely used as both a screening tool in the clinical setting and as an epidemiological tool in many countries [19,20]. The validated Chinese version has been widely used in China. The SDQ comprises five subscales: emotional symptoms, conduct problems, hyperactivity, peer problems, and prosocial behavior. Emotional symptoms and peer problems also form a single “internalizing” subscale, with conduct problems and hyperactivity combining to form a single “externalizing” subscale, with the third subscale, “prosocial behavior”, unchanged. The total difficulties score combined internalizing and externalizing subscales. Higher scores on the total difficulties, internalizing and externalizing subscales represent higher levels of psychological problems, while higher scores on the prosocial behavior subscale represent lower levels of psychological problems. The three-subscale structure of the SDQ, which just includes total, internalizing and externalizing subscales, has been reported to be valid and reduce measurement error. [21]

### 2.4. Data Analysis

Chi-square tests, *t*-tests, and ANOVA were conducted to examine the differences in individual demographic characteristics and SDQ outcomes and self-harming behaviors among four groups of participants. Multiple linear regression and binary logistic regressions models were applied to examine the associations between the psycho-social outcomes and group status, namely, urban, migrant, left-behind, and native-rural children. The SDQ outcomes were included as dependent variables and group status was examined as an independent variable. Analyses were adjusted for age, gender, family economic status, and parents’ education level. All of the analyses were performed while using SPSS 20.0 (Available online: www.esrc.ukri.org/data, accessed 1 July 2019).

## 3. Results

There were 2965 completed questionnaires after excluding children who had been left behind for at least a year, and migrant children who had been living at the destination for at least a year. 57(2%) were excluded for inadequate completion. Table 1 presents the differences in socio-demographic characteristics among participants across the four groups. Significant differences were observed in terms of family economic status and parents’ education level. Most urban children (97%) had high family economic status, as compared with 69% migrant children, 33% rural children, and 24% left-behind children. Most urban children’s parents (79%) had at least high school education, when compared to 31% migrant children’s parents, 22% left-behind children’s parents and 23% rural children’s parents (22.9%).

Table 2 shows the group differences in SDQ-C outcomes. Migrant children reported the highest mean scores for total difficulties (12.5; *p* < 0.001), conduct problems (2.6; *p* < 0.001), peer problems (3.0; *p* < 0.001), internalizing problems (6.5; *p* < 0.001), and externalizing problems (5.0; *p* < 0.001). Left-behind children reported the highest mean scores for emotional symptoms (3.7; *p* < 0.001) and hyperactivity (3.9; *p* < 0.001). Rural children reported lowest mean scores for prosocial behavior (7.1; *p* = 0.001). There were no significant differences between left behind children and migrant children for total difficulties or for any of the individual parameters.

Table 3 shows the linear regression analyses of the SDQ outcomes. After controlling for sex, age, family economic status, and parent’s education level, as compared with urban children, migrant children scored significantly higher for total difficulties (β = 1.50; 95% CI = 0.90, 2.11; *p* < 0.001), internalizing problems (β = 0.94; 95% CI = 0.57, 1.30; *p* < 0.001), and externalizing problems (β = 0.57; 95% CI = 0.20, 0.93; *p* < 0.01), emotional symptoms (β = 0.45; 95% CI = 0.19, 0.70; *p* < 0.01), conduct problems (β = 0.29; 95% CI = 0.10, 0.47; *p* < 0.01), and hyperactivity (β = 0.28; 95% CI = 0.04, 0.52; *p* < 0.05. The left-behind children scored significantly higher for total difficulties (β = 0.92; 95% CI = 0.21, 1.62; *p* < 0.05), internalizing problems (β = 0.59; 95% CI = 0.16, 1.01; *p* < 0.01), and emotional symptoms (β = 0.38; 95% CI = 0.08, 0.67; *p* < 0.05) than urban children did.

## 4. Discussion

As hukou regulations continue to be relaxed in many of China’s cities, it is predicted that more children will accompany their parents to cities, and fewer will be left behind [3]. Our results shed some light on the effects that this might have on children. We found that children living with their parents, in their accustomed environment, that is, urban children and rural children, scored better on the SDQ-C. Urban children scored best for psychological well-being across all parameters with rural children the next best. This result for urban children is perhaps no surprise, given relatively favorable economic circumstances and educational environment, and this has been well documented [16]. However, low socioeconomic status has been found to be a strong predictor of poorer psychological outcomes in Chinese rural children, so the result for the rural children is less expected [12]. But what both groups probably have in common, apart from living with parents in their accustomed environment, is proximity of other relatives, especially grandparents, who often act as primary carers, even when parents are co-residing. We did not ask about this specifically in our questionnaire, but we have conducted some (as yet unpublished) qualitative research, including interviews with children, guardians, parents, and key stakeholders. From county government representatives, we learnt that less than 10% of migrant children had migrated with grandparents. A number of the migrant children at interview specifically mentioned missing grandparents, or said that they wish their grandparents had come with them. The caring role of grandparents has started to gain attention in the literature, with their importance and ubiquity in children’s lives acknowledged [14]. The closeness of child-grandparent relationships, whether parents are present or not, has been observed and documented, especially in rural China [22]. The role of grandmothers in migrant settings has been found in urban migrant setting to play a crucial role in maintaining stability in migrant families’ lives [23]. Accordingly, the absence of this additional family support, especially from grandparents, might contribute to lower well-being scores for migrant children.

The literature generally regards LBC as the most vulnerable group [12], but scores for our migrant children showed greater susceptibility to psychological problems than LBC after adjustment for confounders. LBC and migrant children have consistently similar scores in the SDQ-C, but, after adjustment, migrant children are significantly more likely to be prone to each the four conditions which make-up the total difficulties score. Migrant children score higher for both peer problems and emotional problems than LBC, which suggests that moving with parents to a new environment when parents work long hours and children lack the support of other family members, (as noted above) might make them especially vulnerable. In addition, their lower status in urban society may contribute to more psychological difficulties [5].

We also found unexpected socio-demographic results. Unsurprisingly, children living in urban areas were considerably wealthier (by our measure of wealth by household items) than children in rural areas. However, the finding that left behind children were the poorest is of note. The assumption usually is that LBC are better-off financially than rural children, because of increased income from remittances [24]. However, this seems not to apply here. We hypothesise that rural parents with higher income are choosing not to migrate, or they have already migrated and made enough money to settle back in their hometowns. It is a well-known phenomenon that parents often return to support their children in what are regarded as the more vulnerable years of early adolescence [25], the age group of our participants. In addition, we found that rural parents who take their children are better educated. This might be because better educated parents may see the benefits of exposure to urban lifestyles, culture and opportunity for their children, so-called social remittances. Or this may be related to the specific setting: migrant schools in Hangzhou are mostly above average for migrant schools, even in other parts of Zhejiang, and the education standards are better than in the rural hometown, so they may attract parents, who are more ambitious for their children’s future.

However, our study also highlights methodological issues. It is important to consider the interpretation of the results of the SDQ-C, how the scores compare with normative standards, and what the statistical differences between the scores actually mean. The SDQ-C has been formally validated by Yao et al. [15] and showed high levels of reliability and validity. Accordingly, it is regarded as an appropriate tool for assessing psychopathology in Chinese adolescents. Normative scores for the SDQ-C have also been developed for a predominantly urban sample of adolescents (n = 1135) in Hunan Province. These scores most closely match those of our urban group, with the total difficulties the same: our urban children scoring 10.1 versus the normative 10.2. Overall, while the crude scores of SDQ-C were statistically significant between groups, the range across all of the scores was narrow, and this raises questions regarding clinical significance. For the English version, it is stated that each one point increase in the total difficulties score (out of a total score of 40) corresponds with an increase in the risk of developing a mental health disorder [20]. But a total difficulties score of >14 is only “slightly raised”. The highest mean total difficulty score in our study was 12.5 (SD 5.2) for migrant children with 12.3 (SD 5.1) for left behind children, so the majority of children fall within normal limits. Hence, the differences are small and the interpretation of statistically significant differences needs to be treated with caution.

There are clear limitations. The study was conducted in one province, with a relatively small sample size in each age group. We used just one measure of psychological well-being, which does not allow for triangulation within the study, although our qualitative component will help to fill that gap. Only children with parental or guardian consent to inclusion were able to participate, which may have biased the samples. The use of schools as the sampling frame excluded the very small number of children not attending school. This would include children with disabilities and mental health problems, who may be especially vulnerable to the effects of parental migration. Further research should explore this area. In addition, the clustering effects of the use of schools have not been considered. Self-report, especially in children and in relation to sensitive issues, might lead to reporting bias. We did not ask specifically about proximity of extended family members, and have had to make inferences based on norms. As with all cross-sectional methodology, it does not infer causation. Our concerns about the real meaning of small but statistically significant differences in SDQ-C scores are outlined above.

## 5. Conclusions

There are small but significant differences in total difficulties across the four groups of children categorized by migration and hukou status. Migrant children and left behind children, the two groups that are affected by migration, are the most vulnerable. Urban and rural children score best, despite the socioeconomic circumstances of the latter. Our results suggest that children do best when living with their parents in their accustomed environment, and probably when extended family are nearby. Migrant children fare less well, partly because of displacement, because parents are working very long hours, and perhaps because of the absence of extended family. While the migrant schools in our study were reasonably good for this type of school, this is not the case in many cities, and the government is committed to addressing this. The State Council recently announced a policy that specifically mentions the educational enrolment of migrant children, stating that they should enjoy equal opportunity in education, and that at least 85% of migrant children should be educated in state schools [8]. The relaxation in hukou rules and the encouragement of rural hukou holders to settle in small and medium sized cities, where there is an increasing need for labour and talent, is going to result in many more children staying with their parents and fewer left behind children. Accordingly, the education policy is very timely. Migration to cities with parents and possibly other family members, especially grandparents, may also mitigate against the other big challenge for the Chinese government of left behind elderly in rural areas [26].

## Figures and Tables

**Table 1 ijerph-16-03744-t001:** The socio-demographic characteristics of participants.

Socio-Demographic Variables	Overall n = 2965 n (%)	Urban Children n = 867 n (%)	Migrant Children n = 625 n (%)	Left-Behind Children n = 778 n (%)	Rural Children n = 695 n (%)	*χ*^2^ or F	*p* Value
Gender						10.86	0.013
Male	1580 (53)	454 (52)	350 (56)	382 (49)	394 (57)		
Female	1385 (47)	413 (48)	275 (44)	396 (51)	301 (43)		
Age, M(SD)	12.02 (2.19)	11.86 (1.19)	11.97 (1.47)	12.84 (2.31)	13.52 (2.27)	1056.6	<0.001
Family economic status						1139.4	<0.001
Poor	428 (15)	1 (0.11)	47 (7.5)	229 (29)	151 (22)		
Moderate	842 (28)	22 (2.5)	145 (23)	364 (47)	311 (45)		
Wealthy	1695 (57)	844 (97)	433 (70)	185 (24)	233 (33)		
Parents’ education level						1124.7	<0.001
Illiterate or primary School	617(21)	63(7.3)	83(13.3)	258(33)	213(31)		
Middle school	1142(39)	124(14)	350(56)	345(44)	323(46)		
High school	664(22)	239(28)	160(26)	133(17)	132(19)		
College or above	542(18)	441(51)	32(5.1)	42 (5.4)	27(3.9)		

**Table 2 ijerph-16-03744-t002:** Group differences in self-reported version of the Strengths and Difficulties Questionnaire (SDQ-C) outcomes.

SDQ Outcomes	Urban Children Mean (SD)	Rural Children Mean (SD)	Migrant Children Mean (SD)	Left-Behind Children Mean (SD)	*p* Value LBC *vs* Migrants	*p* Value All Four Groups
**Total difficulties**	10.1 (5.4)	11.9 (4.8)	12.5 (5.2)	12.3 (5.1)	0.09	<0.001
**Emotional symptoms**	2.9 (2.1)	3.4 (2.1)	3.5 (2.1)	3.7 (2.2)	0.2	<0.001
**Conduct Problems**	2.0 (1.6)	2.1 (1.5)	2.4 (1.6)	2.1 (1.4)	0.07	<0.001
**Hyperactivity**	3.0 (2.2)	3.8 (1.9)	3.7 (2.0)	3.8 (2.1)	0.3	<0.001
**Peer problems**	2.3 (1.7)	2.6 (1.6)	3.0 (1.8)	2.7 (1.7)	0.06	<0.001
**Prosocial behavior**	7.5 (2.1)	7.2 (1.9)	7.3 (1.9)	7.2 (2.0)	0.3	0.001
**Internalizing problems**	5.2 (3.1)	6.0 (3.0)	6.5 (3.2)	6.4 (3.1)	0.4	<0.001
**Externalizing problems**	5.0 (3.3)	5.9 (2.9)	6.1 (3.0)	5.9 (3.1)	0.5	<0.001

**Table 3 ijerph-16-03744-t003:** Regression coefficients for SDQ-C outcomes after adjustment for sociodemographic characteristics.

Migration Status	Total Difficulties β(95%CI)	Internalizing Problems β(95%CI)	Externalizing Problems β(95%CI)	Emotional Symptoms β(95%CI)	Conduct Problems β(95%CI)	Hyperactivity β(95%CI)	Peer Problem β(95%CI)	Prosocial Behavior β(95%CI)
Urban children	1.00	1	1	1.00	1.00	1.00	1.00	1.00
Migrant children	1.50 (0.90, 2.11) ***	0.94 (0.57, 1.30) ***	0.57 (0.20, 0.93) **	0.45 (0.19, 0.70) **	0.29 (0.10, 0.47) **	0.28 (0.04, 0.52) *	0.50 (0.29, 0.69) ***	0.19 (−0.04, 0.42)
Left-behind children	0.92 (0.21, 1.62) *	0.59 (0.16, 1.01) **	0.33 (−0.10, 0.76)	0.38 (0.08,0.67) *	0.11 (−0.11, 0.32)	0.22 (−0.06, 0.51)	0.21 (−0.02, 0.44)	0.11 (−0.16, 0.38)
Rural children	0.47 (−0.26, 1.19)	0.15 (−0.28, 0.59)	0.31 (−0.12, 0.75)	0.04 (−0.26, 0.33)	0.13 (−0.09, 0.35)	0.18 (−0.11, 0.47)	0.12 (−0.12, 0.36)	0.03 (−0.25, 0.31)
Sex								
Male	1.00	1	1	1.00	1.00	1.00	1.00	1.00
Female	−0.23 (−0.67, 0.22)	0.01 (−0.26, 0.27)	−0.23 (−0.50, 0.03)	0.37 (0.19, 0.56) ***	−0.15 (−0.29, −0.02) *	−0.08 (−0.26, 0.10)	−0.37 (−0.51, −0.22)	0.47 (0.30, 0.64) ***
Age	0.04 (−0.08, 0.16)	0.02 (−0.05, 0.09)	0.02 (−0.06, 0.09)	0.07 (0.02, 0.12) **	−0.03 (−0.07, 0.01)	0.05 (−0.01, 0.09)	−0.05 (−0.09, −0.01) *	0.07 (0.02, 0.12) **
Economic status								
Poor	1.00	1	1	1.00	1.00	1.00	1.00	1.00
Moderate	−0.21 (−0.81, 0.16)	−0.16 (−0.52, 0.20)	−0.05 (−0.41, 0.31)	−0.03 (−0.28, 0.22)	0.09 (−0.09, 0.27)	−0.14 (−0.38, 0.10)	−0.13 (−0.33, 0.07)	0.02 (−0.21, 0.25)
Wealthy	−0.73 (−1.36, −0.10) *	−0.53 (−0.91, −0.15) **	−0.20 (−0.58, 0.18)	−0.24 (−0.50, 0.02)	0.05 (−0.14, 0.24)	−0.25 (−0.50, 0.01)	−0.29 (−0.50, −0.09)	0.37 (0.12, 0.61) **
Parents’ education								
Illiteracy or Primary	1.00	1	1	1.00	1.00	1.00	1.00	1.00
Middle school	−0.02 (−0.53, 0.49)	−0.30 (−0.60, 0.01)	0.28 (−0.03, 0.58)	−0.07 (−0.28, 0.14)	0.01 (−0.14, 0.16)	0.27 (0.06, 0.47) **	−0.23 (−0.40, −0.06) **	0.12 (−0.07, 0.32)
High school	−0.22 (−0.80, 0.37)	−0.03 (−0.38, 0.32)	−0.19 (−0.54, 0.16)	0.08 (−0.16, 0.32)	−0.02 (−0.20, 0.16)	−0.17 (−0.40, 0.07)	−0.11 (−0.30, 0.09)	0.49 (0.26, 0.71) ***
College or above	−1.64 (−2.34, −0.95) ***	−0.83 (−1.25, −0.41) ***	−0.81 (−1.23, −0.39) ***	−0.39 (−0.68, −0.10) **	−0.18 (−0.38, 0.03)	−0.64(−0.91, 0.36) ***	−0.44 (−0.67, −0.21)	0.72 (0.45, 0.98) ***

**p* < 0.05, ***p* < 0.01, ****p* < 0.001.

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
