# Peer review of "Relaxation in the Chinese Hukou System: Effects on Psychosocial Wellbeing of Children Affected by Migration"

_ijerph, 2019, doi:10.3390/ijerph16193744_

Round 1
Reviewer 1 Report
This manuscript conducts cross-sectional, school-based survey using the Chinese version of the Strengths and Difficulties Questionnaire (SDQ) in urban and migrant schools. The sample included 867 urban children with urban hukou, 625 migrant children with rural hukou, 695 rural children with rural hukou living with parents, and 778 left behind children. The results showed that children directly affected by migration were most at risk of psychological problems; urban and rural children living with their parents in their accustomed environment, were least vulnerable.
The issue of why SDQ score is different for different children group is important, but several issues would need to be addressed to make this a useful contribution to the literature.
Only small part variables are included in multivariate models to control, other variables, such as the number of siblings, whether the child is the boarding students, are high correlating with the SDQ score. If these variables were not been controlled, the models may not be accurately specified. It is important to consider clustering of students within schools even within classes. Is there evidence that the SDQ is valid for Chinese children? The authors should avoid causal language given that their data are cross-section.Author Response
Reviewer 1 -
Thank you very much for the comments. Our response is in bold below.
Only small part variables are included in multivariate models to control, other variables, such as the number of siblings, whether the child is the boarding students, are high correlating with the SDQ score. If these variables were not been controlled, the models may not be accurately specified.
We deliberately analysed the major socio-demographic variables for this comparison of four groups. More complex models would have become more difficult to interpret accurately. Eg there were no children in boarding school in urban areas. In rural areas boarders were only in middle school; 62% of children in middle school were boarding. We did preliminary analysis of SDQ scores in boarders and non-boarders and found no significant differences. We therefore did not use “boarding” as a predictor variable in comparing these four groups.
It is important to consider clustering of students within schools even within classes.
Yes we agree to some extent, though the effects in this context are unlikely to affect the overall result and clustering is of course of most importance in a trial setting. But we have added this to the limitations section.
Is there evidence that the SDQ is valid for Chinese children?
Yes. As stated (Line 133) the “The validated Chinese version has been widely used in China” and lines 230-231 “The SDQ-C has been formally validated by Yao et al [15] and showed high levels of reliability and validity”
The authors should avoid causal language given that their data are cross-section.
We do. But have added the caveat to the limitations.
Reviewer 2 Report
As a pediatrician living in China for 10 years and interacting with high risk children such as those described in this study and facilitating mental health support as well, I am very interested in this study. Thank you for looking more intently at this very important group of children. I would like to see some comments about perhaps the highest, most invisible group of migrant children-those unable to attend school for various reason (hukou not good in the city, or they are in villages where the schools are too far away). The fact that all the children were enrolled in school makes that an automatic inclusion criteria. Can any information be gathered about numbers of children in the communitites studied who might not be in school for various reasons? What about children who are refused school attendance because of mental health/behavior problems that have gotten them expelled or makes it too difficult to attend? there may be a significant population not addressed in this study that could be focused on in future projects. Thank you for sharing your research!
Author Response
Reviewer 2
Thank you very much for the comment below from Reviewer 2.
“I would like to see some comments about perhaps the highest, most invisible group of migrant children-those unable to attend school for various reason (hukou not good in the city, or they are in villages where the schools are too far away). The fact that all the children were enrolled in school makes that an automatic inclusion criteria. Can any information be gathered about numbers of children in the communities studied who might not be in school for various reasons? What about children who are refused school attendance because of mental health/behavior problems that have gotten them expelled or makes it too difficult to attend?”
We totally agree that there is a small, but very important, group of children who do not attend school, and who by definition are excluded from this study. We have added this to the limitations (Page 7, Para 2). These vulnerable children are of course harder to access, and it would be hard to make appropriate comparisons with non-vulnerable children. But the effects of migration on these children would be a very interesting area for us to research in the future. So we really thank the reviewer for raising this.

Reviewer 3 Report
Overall, this is a quite important and valuable research paper concerning the impact of urban migration on the mental health of children.
A few general suggestions:
It might be good for the reader who may be unfamiliar with the hukou system to mention how this system perpetuates particular social attitudes and biases urban residents often hold for rural residents and how these biases against, and exclusions for, rural migrants are historical and structural.
These biases and the stresses that both migrant parents and children may experience in urban areas may also help to explain the results for migrant children compared to left-behind children.
Section 2.1 Sample selection: According to Table 1 the sexes of the sample appear fairly balanced...was this purposive in the sample selection process?
Section 2.2 What happened to the raw data after data analysis? Was it destroyed?
Some particular suggestions:
Line 56: can you provide some examples of some of these 'smaller' cities?
Line 57: 'This all...' can you be more specific to what you are referring?
Line 66: I question if the paragraph starting at this line can be reworked to flow better from one idea to the next? For example, Line 76 'Hukou restrictions...' does not seem to logically follow from what came before. It possibly just needs a bridge from what came before...
Line 101: 'the SDQ is regarded as an especially appropriate tool...' can you either include a citation or identify by whom is this tool regarded to be appropriate. Further down lines 131-133 you explain this, but maybe this should be moved up to line 101 when you first mention the SDQ.
Line 153: it might be helpful to rephrase the following sentence which is now a bit confusing: '...who had not been in the situation for at least one year'.
Author Response
Many thanks to the reviewer for the constructive comments.
A few general suggestions:
It might be good for the reader who may be unfamiliar with the hukou system to mention how this system perpetuates particular social attitudes and biases urban residents often hold for rural residents and how these biases against, and exclusions for, rural migrants are historical and structural.
We appreciate this. We think that a detailed description of the hukou system is unnecessary, since references are provided for this. However, we have added one extra sentence which refers to the biases against rural residents which are perpetuated by the hukou system. (Page 1, Para 3)
These biases and the stresses that both migrant parents and children may experience in urban areas may also help to explain the results for migrant children compared to left-behind children.
We have added a note in the discussion to this effect. (Page 6, Para 2)
Section 2.1 Sample selection: According to Table 1 the sexes of the sample appear fairly balanced...was this purposive in the sample selection process?
No. It just happened this way.
Section 2.2 What happened to the raw data after data analysis? Was it destroyed?
Raw data is securely stored. Under university rules we have to keep raw data for 10 years.
Some particular suggestions:
Line 56: can you provide some examples of some of these 'smaller' cities? Added
Line 57: 'This all...' can you be more specific to what you are referring? Added
Line 66: I question if the paragraph starting at this line can be reworked to flow better from one idea to the next? For example, Line 76 'Hukou restrictions...' does not seem to logically follow from what came before. It possibly just needs a bridge from what came before...
We disagree. The flow is good. The sentence “Hukou restrictions…” expands the point about the reason for the difficulties of migrant children.
Line 101: 'the SDQ is regarded as an especially appropriate tool...' can you either include a citation or identify by whom is this tool regarded to be appropriate. Further down lines 131-133 you explain this, but maybe this should be moved up to line 101 when you first mention the SDQ.
Thank you. This was an obvious omission. We have added a reference for statement [18].
Line 153: it might be helpful to rephrase the following sentence which is now a bit confusing: '...who had not been in the situation for at least one year'.
We agree. We have changed this
Round 2
Reviewer 1 Report
I did not see the attempt to improve the paper. The reply was very casual and lacking evidence.
1) The boarding students are similar as the LBCs, due to that they also did not live with their family in most of the time. And in rural schools, especially in middle schools, the boarding rate is very high. And it is not like the authors claimed there is no boarding student in urban areas. Further, in the response letter, it is said:” We did preliminary analysis of SDQ scores in boarders and non-boarders and found no significant differences.” But there is no table or data show the result in the updated manuscript. And they did not response for the sibling issue.
2) If they have the data, it is easy to add the school fixed effect or class fixed effect. I did not know why they do not want to do this.
3) For the causal effect, at least they can do the propensity score matching to try to control for the endogeneity issue. But they did not.